# Microphones as Airspeed Sensors for Unmanned Aerial Vehicles

**DOI:** 10.3390/s23052463

**Published:** 2023-02-23

**Authors:** Momchil Makaveev, Mirjam Snellen, Ewoud J. J. Smeur

**Affiliations:** Faculty of Aerospace Engineering, Delft University of Technology, Kluyverweg 1, 2629 HS Delft, The Netherlands

**Keywords:** tail-sitter, airspeed, hydrodynamic pressure fluctuations, pseudo-sound, turbulent boundary layer, microphones, power spectral density, feed-forward neural networks

## Abstract

This paper puts forward a novel design for an airspeed instrument aimed at small fixed-wing tail-sitter unmanned aerial vehicles. The working principle is to relate the power spectra of the wall-pressure fluctuations beneath the turbulent boundary layer present over the vehicle’s body in flight to its airspeed. The instrument consists of two microphones; one flush-mounted on the vehicle’s nose cone, which captures the pseudo-sound caused by the turbulent boundary layer, and a micro-controller that processes the signals and computes the airspeed. A feed-forward single-layer neural network is used to predict the airspeed based on the power spectra of the microphones’ signals. The neural network is trained using data obtained from wind tunnel and flight experiments. Several neural networks were trained and validated using only flight data, with the best one achieving a mean approximation error of 0.043 m/s and having a standard deviation of 1.039 m/s. The angle of attack has a significant impact on the measurement, but if the angle of attack is known, the airspeed could still be successfully predicted for a wide range of angles of attack.

## 1. Introduction

Airspeed indicators were used in manned aircraft as early as 1912 [1]. Airspeed measurement is crucial for pilots to guarantee a safe and effective flight. Likewise, for Micro Air Vehicles (MAVs), in particular those with a wing, airspeed is an important variable. The aerodynamic forces and moments act on the MAV scale with the airspeed squared [2], and so do the forces and moments that are generated with the control surfaces. This makes airspeed a crucial input variable for an automatic flight control system to achieve precise and safe flight control.

Traditionally, Pitot tubes have been used in aircraft to measure the vehicle’s airspeed. They are mechanically simple and have been used with success for decades. However, there are some problems with Pitot tubes. First, as the measured dynamic pressure is proportional to the airspeed squared, the accuracy of a Pitot tube decreases as the airspeed decreases [3]. Second, in order to provide accurate measurements, a Pitot tube needs to be aligned with the direction of the airflow and cannot have a large incidence angle. This is especially problematic for slow-flying MAVs of the hybrid tail-sitter type, as they combine both hovering and forward flight capabilities, tilting themselves forward to transition from vertical to horizontal flight [4]. These vehicles may encounter very large angles of attack, causing the Pitot tube to be misaligned with the airflow, leading to a reduced dynamic pressure measurement and potentially an increased static pressure measurement, overall producing a lower measured airspeed.

There are alternatives to the traditional Pitot-static tube. Hayward [5] lists many different instruments for flow measurement, but most are designed to measure the flow in pipes and are not really applicable for installation on an MAV. Instruments that could be installed on aircraft are the vane-type anemometer and hot-wire anemometer. The vane-type anemometer is basically a miniature windmill, which measures the flow velocity by measuring the rotational speed of the rotor. It can be accurate but generates drag, and it suffers from the same problem as the Pitot tube in that it needs to be aligned with the flow.

The hot-wire anemometer, or the variant where the hot-wire is replaced with a thermistor, works by keeping the hot element at an elevated temperature. The more airflow that passes the hot element, the more power that is needed to maintain the element’s temperature, which provides a measure for the velocity. The benefit of this sensor is that it can be made omnidirectional to some degree and can be accurate at low velocities [6]. A downside is that it is rather fragile and requires precise calibration.

Alternatively, one can derive the airspeed from a combination of a model of the vehicle dynamics and inertial sensors and GPS [7]. Sun and Gebre-Egziabher [8] use a Kalman filter to estimate the airspeed. However, this method requires an accurate model of the vehicle, and the GPS signal may not always be available. As such, there is an existing need for an alternative airspeed sensor for small UAVs—one that can operate in the different flight modes of the vehicle, at any time and geographical location, without requiring frequent maintenance.

The literature suggests there might be an alternative method to measure airspeed using microphones. Air flowing over an aircraft’s skin forms a Turbulent Boundary Layer (TBL), and the airspeed affects the spectral features of the pressure fluctuations in this boundary layer [9,10], which can be measured with a microphone as pseudo-sound [11]. Smol’yakov [12] and Rackl’s [13] work in modeling these spectra is aimed at noise control, especially in aircraft cabins, while Laganelli [14] and Lowson [15] also consider the prospect for fatigue failure analysis and prediction of the aircraft’s structure. Edelman et al. [16] utilize spectral features to detect boundary layer separation. In all of these models aimed at predicting the TBL-induced surface pressure fluctuations, the airspeed is an independent variable. While Moshkov [17] elaborates on the influence of the Mach number on the surface pressure fluctuation levels measured on the fuselage of a jet in subsonic flight and Panton [18] examines the pressure–velocity correlations for data from microphones mounted on the fuselage of a sailplane, no dedicated models with the airspeed as the dependent variable are put forward.

In this paper, we propose and evaluate a novel design for an airspeed sensor based on microphones placed in the fuselage skin of an MAV. The underlying working principle of the sensor is to relate the Power Spectral Density (PSD) of the wall-pressure fluctuations in the TBL to the vehicle’s airspeed. Wind tunnel and flight experiment data are used to train a neural network to predict the airspeed based on the shape and magnitude of the PSD. The Cyclone [19] MAV shown in Figure 1 is used as a development and testing platform for the proposed airspeed instrument due to its configuration and the available space inside its fuselage. It is a 1.1 kg tail-sitter fixed-wing MAV that has the most efficient cruise speed at 16 m/s and can fly up to a speed of 30 m/s.

## 2. Design of the Airspeed Instrument

### 2.1. Microphone Selection and Configuration

The microphones should be small, light, and low power in order to minimize their impact on the MAV performance. A flat frequency response curve over the frequencies of interest is desired as it means that the signals from the microphones will better represent the true pseudo-sound caused by the TBL. For the same reason, a low self-noise level is desired. Finally, the directionality of the microphones needs to be considered. In experiments measuring the hydrodynamic pressure fluctuations in the TBL, omnidirectional microphones are typically used [11]. The reason is that the turbulent flow structures are of three-dimensional nature, meaning that hydrodynamic pressure fluctuations propagate from all directions around the microphone [20].

Based on the desired characteristics and performance of the microphones, the Sonion 8002 is selected. It is an omnidirectional analog electret microphone with a cylindrical shape, having a diameter of 2.5 mm, a height of 3 mm, and a total weight of less than 1 g. The sensitivity is −33.5 dB*_SPL_* at 1 kHz relative to 1 V/Pa. The sensitivity is quite flat over the frequency range of 1 to 10 kHz, with a slight sensitivity increase at the higher frequencies; the sensitivity at 10 kHz is −30 dB*_SPL_*. The sensitivity curve from the microphone data-sheet is used to obtain unbiased spectrograms (https://www.sonion.com/product/8002-2/, (accessed on 10 January 2023)). A benefit of electret microphones is that the diaphragm is not attached to a coil, meaning that it can vibrate more freely, thus providing a more precise representation of the dynamic pressure waves hitting the microphones [21]. A disadvantage of the electret microphones is the presence of active components, which makes moisture problematic, thus limiting the usability of the airspeed sensor in rainy weather.

The microphones are mounted flush with the vehicle’s skin, as shown in Figure 2. Having the microphones protruding from any surface of the vehicle could distort the boundary layer flow over that particular part of its body, thus impacting the spectral features of the pseudo-sound [22], while mounting the microphones in a cavity under the skin is found to attenuate the captured pseudo-sound [23]. The mounting location of the microphones on the vehicle’s body is also of importance, as towards the front of the vehicle, laminar flow may be found, while towards the rear, the boundary layer may be more turbulent or even separated. For the Cyclone, the nose cone is chosen as the most suitable location to mount the microphones. To make sure the boundary layer at the microphones is turbulent, a 60° zig-zag turbulator tape with a thickness of 0.5 mm is placed 30 mm upstream of the microphones’ mounting location. The microphones themselves are mounted 55 mm downstream of the nose cone’s leading edge and are held in place solely by the tight fit in the mounting hole.

A potential problem that we address here is that the microphones may also capture sounds from the surrounding environment, most notably from the motors and the propellers. The presence of such sounds may influence the airspeed measurement. A technique to isolate the component of the hydrodynamic pressure fluctuations is to subtract the signals of two microphones mounted at distinct locations [11,22]. This requires the assumption that the sound is in phase, which would be the case for sound originating from somewhere in the symmetry plane of the aircraft. This method is based on the fact that the hydrodynamic pressure fluctuations at different points on the surface should be uncorrelated [24]. Generally, the pressure fluctuations captured by the microphones can be contributed to three distinct sources, namely acoustic sound, TBL-induced hydrodynamic pressure fluctuations and vibrations of the surface itself [22]:
(1)$$p^{\prime }=p_{a}^{\prime }+p_{TBL}^{\prime }+p_{v}^{\prime }$$

The symbol $p^{\prime }$ denotes the fluctuating component of the pressure, $p_{a}^{\prime }$ refers to the contribution of the acoustic sound, $p_{TBL}^{\prime }$ to the contribution of the pseudo-sound, and $p_{v}^{\prime }$ to the pressure fluctuations caused by surface vibrations. Given that definition, if the subtraction method is applied to two separate microphones, the following holds for the mean of squares of the pressure fluctuations:(2)$$\bar{{\left(p_{1}^{\prime }-p_{2}^{\prime }\right)}^{2}}=\bar{{\left(p_{a_{1}}^{\prime }+p_{TBL_{1}}^{\prime }+p_{v_{1}}^{\prime }-p_{a_{2}}^{\prime }-p_{TBL_{2}}^{\prime }-p_{v_{2}}^{\prime }\right)}^{2}}$$

Assuming that the microphones are mounted on the same rigid surface, the mean vibrational component $\bar{p_{v}^{\prime }}$ can be canceled. Furthermore, assuming that the acoustic components of the pressure fluctuations are in-phase, they will be suppressed when taking their difference. Then, expanding Equation (Equation 2) and setting products of uncorrelated coherent components to zero [22], it can be found that the variance in the subtracted signal depends only on the variance of the TBL pressure:
(3)$$\bar{{\left(p_{1}^{\prime }-p_{2}^{\prime }\right)}^{2}}=2\bar{{\left(p_{TBL}^{\prime }\right)}^{2}}$$

For flush-mounted microphones beneath a TBL, studies in wind tunnels have found that for the same streamwise location, a distance *x* past the leading edge, separation of at least one boundary layer thickness δ is needed between the microphones in order for any correlation between their signals to be attributed to the acoustic sounds in the environment [25]. As such, it is decided to have two microphones as part of the airspeed instrument. The two microphones are spaced out at 40 mm apart, which is more than the expected boundary layer thickness of 2.2 mm at their mounting location, estimated from theory for incompressible TBL over a flat plate, where the Reynolds number is taken for airflow at standard sea level conditions and a free-stream velocity of 20 m/s [26]:
(4)δ=0.37xRex1/51+Rex6.9×10721/10

However, it should be noted that the assumption that the acoustic components of the pressure fluctuations are perfectly in-phase is valid only for acoustic sounds whose wave fronts are perpendicular to the vehicle’s longitudinal axis. The phase characteristics of the microphones, if different, may also affect the performance of the subtraction filter.

### 2.2. Supporting Components and Configuration of the Instrument Board

With the goal of recording microphone data in-flight and the end goal in mind of developing an instrument that can measure airspeed in real-time using microphones, hardware components are selected. Important drivers in the component selection were availability, ease of implementation, weight and size.

As processing audio in real-time requires quite some computational power, a dedicated micro-controller is implemented as part of the airspeed instrument. The Teensy^®^ 4.1 (https://www.pjrc.com/store/teensy41.html (accessed on 10 January 2023)) development board was selected, which incorporates an ARM Cortex-M7 processor. The airspeed instrument was completed by an Analog to Digital Converter (ADC), a dedicated battery, a linear voltage regulator and an on/off switch.

An external 16-bit TI PCM1808 (https://www.ti.com/lit/ds/symlink/pcm1808.pdf (accessed on 10 January 2023)) ADC is included as the Teensy^®^ 4.1’s internal ADC and cannot reach the required sampling frequency (https://www.arduino.cc/reference/en/language/functions/analog-io/analogread/ (accessed on 10 January 2023)). Having a dedicated battery makes it easy to install the instrument in different vehicles and avoids noise from fluctuations in the drone’s power supply. Lastly, the linear voltage regulator brings the battery voltage from 7.4 to 5 V to power the micro-controller. Figure 3 depicts the block diagram of the instrument. The instrument board is mounted in a box that houses the battery directly below the microphones in the nose one. It has a total weight of 71 g, length of 87 mm, width of 57 mm and height of 33 mm, taking into account the battery housing box and a total power consumption of up to 120 mA at 5V.

## 3. Experiments

Single-point frequency spectrum models have been described that aim to predict the flow-induced wall-pressure fluctuations in a TBL [27]. However, those models are semi-empirical, meaning that they have been—at least to some extent—developed using data from experiments. Furthermore, different models are better suited for distinct flow conditions and regimes [26]. Therefore, it is beneficial to perform dedicated experiments to gather data about the specific wall-pressure fluctuations induced by the flow structures in the TBL at the microphones’ mounting location on the vehicle’s body and study their dependence on the airspeed and the Angle of Attack (AoA) using the same microphones and processing techniques as those that are part of the airspeed instrument. The experiments include both wind tunnel and flight tests. The wind tunnel experiments can be well-controlled, making them useful in understanding the underlying relations between the spectrum of the wall-pressure fluctuations and the airspeed for a wide range of AoA, while the flight experiments provide data representative of real flight affected by environmental factors.

### 3.1. Wind Tunnel Experiments

The wind tunnel experiments are carried out at the A-Tunnel of the TU Delft’s Low-Speed Laboratory, shown in Figure 4. It is a vertical low turbulence tunnel with an open test section, where the flow comes through a circular exit with a diameter of 0.6 m. The turbulence level of the airflow is below 0.1%, achievable due to the high contraction ratio of the settling chamber. It is a semi-anechoic aeroacoustic wind tunnel with low operating noise, making it ideal for the planned experiments. The wind tunnel is assumed to be anechoic above 200 Hz [28].

The wind tunnel experiments are characterized by three independent variables: the airflow velocity (measured in meters per second), the angle of attack (measured in degrees) and the UAV’s motor configuration. The last independent variable has the following states defined: propellers dismounted (D), propellers mounted but motors are turned off (M), and mounted propellers with motors running at 30% throttle (R).There is only one dependent variable defined for the wind tunnel experiments, namely the microphones’ voltage signals. Each experimental run consists of 15 s of recorded signals. It should be noted that high angles of attack are not tested at high airspeeds as this is not a flight condition of interest since the higher angles of attack are encountered during the hybrid UAV’s transition to a vertical hover, which usually takes place at lower airspeeds. Figure 5 depicts the experiment matrix for the wind tunnel experiments. The matrix is valid for all three motor configurations of the UAV.

To ease the data acquisition process, during the wind tunnel experiments, a National Instruments CompactRIO is utilized instead of the instrument’s hardware to acquire the microphones’ voltage signals. The acquisition system has a 24-bit ADCs with a sampling frequency of 51.2 kHz (https://www.ni.com/en-rs/support/documentation/supplemental/17/specifications-explained--c-series-modules.html#ADCresolution (accessed on 10 January 2023)).

### 3.2. Flight Experiments

The flight experiments are manually piloted flights, conducted outside with isolated phases of steady forward flight at the different airspeed values of interest. Two flights are performed, with a duration of 13 and 15 min.

The independent variables of the flight experiments are the vehicle’s airspeed, the AoA and the motors’ Revolutions per Minute (RPM). A ground truth for the airspeed is provided with a Pitot tube connected to the autopilot board. Next to that, a GPS sensor is present that provides data about the vehicle’s ground speed vector, but the UAV lacks an AoA sensor. The dependent variable is the microphones’ voltage signals. A single measurement of the dependent variable corresponds to a microphone recording of a steady forward flight phase with a length of 1 s at a particular airspeed.

It should be noted that due to environmental factors, such as wind gusts, the airspeed is not constant over the whole forward flight phase. Therefore, an allowed range of 0.5 m/s is defined. As such, if the airspeed as registered by the Pitot tube within one second of data has a range less than 0.5 m/s, it is accepted as corresponding to a steady forward flight phase with the airspeed value taken to be the average over the 1 s measurement.

No experiment matrix is defined for the flight tests as only the airspeed is varied over its defined range, while the corresponding motor RPM for a given airspeed might differ due to wind gusts. The airspeed values recorded in the steady forward flight phases start from as low as 6 m/s and reach up to 20 m/s. The microphone and autopilot data are logged by different devices, each having its own clock. The data are synchronized by identifying the moment of starting and stopping the motors before and after the flight in both logs.

## 4. Data Processing

For the wind tunnel data, a window of fixed length of 1 s is defined, which extracts the microphone signal within its boundaries. The window moves over the recording by 0.1 s each step. A window of the same length is used when extracting flight data, but it moves over the airspeed data logged to the autopilot by the Pitot tube. At each step, both the average and the range of the airspeed data in the window are calculated, and given that the minimum airspeed and data range requirements are met, the corresponding microphones’ signals for that time segment are extracted and saved.

The microphones’ signals collected from the wind tunnel experiments are dominated by a 50 Hz component. This is the case for all the data collected during the wind tunnel experiments, as well as the ones where the wind tunnel was not running. The most likely conclusion is that the 50 Hz component comes from the electrical sockets, which are 50 Hz Alternating Current (AC). This is further substantiated by the fact that the 50 Hz component is not present in the flight data, where a battery is used to power the microphones. This issue is resolved by passing the microphones’ signals through a second-order high-pass filter, with the cut-off frequency set at 250 Hz, chosen to be also higher than the cut-off frequency of the wind tunnel’s semi-anechoic chamber. Figure 6 depicts an example of the resulting time-domain signals, taken for the left-mounted microphone, upon applying the high-pass filter.

The next step in the data processing is to compute the PSD for all extracted segments of microphones’ signals, both from the wind tunnel and flight experiments. To do so, Welch’s method is utilized, where the PSD estimation is performed by dividing the time signal into successive segments, forming a periodogram for each segment and then averaging. The reason behind choosing this approach is that the variance of the periodogram is reduced by breaking the time series signal into segments that typically overlap [29]. The window function used as part of Welch’s estimation is a Hamming window of length M= 1024 samples. As the number of samples in the window is an even number, the overlap between adjacent windows is set to 50% as recommended for Hamming windows, resulting in a window hop size of R= 512 samples [30].

The frequency range of interest is bounded on the lower side by the high-pass filter’s cut-off frequency, while the upper bound is at 14 kHz as, for higher frequencies, the PSDs do not vary with airspeed. Within that range, it was found that working with the frequencies between 1 and 10 kHz yielded models with comparable accuracy in predicting the airspeed while needing fewer sample points from the PSD function. The resulting estimates obtained using Welch’s method for the wind tunnel and flight microphone signal segments can be seen in Figure 7 for both the left and right microphones. The plots shown in orange correspond to the PSD of the wind tunnel microphone data segment, while the plots in blue correspond to the flight microphone data segment. It can be seen that the flight data generally follow the same pattern as the wind tunnel data, though it has a bit more power in the low frequencies, while having less power at the high frequencies. The difference could potentially be attributed to the different ADCs used, the influence of the mounting stand in the wind tunnel, the possibility of small amounts of sideslip in the real test flight and different thrust levels given in the real flight. The magnitude of the PSD is expressed in dB*_SPL_* by converting the microphones’ voltage signals to pressure signals in Pa using the microphones’ sensitivity data. The highest sound pressure level (SPL) of a recording is found to be 98.4 dB*_SPL_*, corresponding to the right microphone during the wind tunnel experiments for motor configuration R and an airspeed of 10 m/s at an AoA of 45°. This is below the microphone’s maximum input level, so there should not be any distortions present in the recorded signals.

The signal subtraction method is applied directly to the microphones’ voltage signals in the time domain as the recorded microphone voltage is related to the corresponding magnitude of the pressure fluctuations captured by the microphone only by a scaling factor determined by the microphone’s sensitivity:
(5)$$V_{sub}=V_{left}-V_{right}$$

The $V_{sub}$ signal is treated in the same manner as the ΔV signals from the microphones, with the PSD computed using Welch’s method, and its magnitude subsequently expressed in dB*_SPL_*. Figure 8 depicts the PSDs of the $V_{sub}$ signals for both the wind tunnel and flight microphone data segments. From the plots, it can be seen that the PSD of the resulting signal from the wind tunnel microphone data has less energy in the frequencies below 2 kHz compared to its flight data equivalent. Next to that, there is a spike at 8.7 kHz present only in the wind tunnel data. This behavior for the wind tunnel PSDs is observed in all experimental trials. Its lower PSD magnitude in the low-frequency region could point towards stronger turbulent structures of lower convection velocity forming in the TBL during flight, while the peak at 8.7 kHz could be attributed to a harmonic of the 50 Hz component from the AC as it is present in all PSDs of the subtracted microphones’ signals collected during the wind tunnel experiments, even for the scenario when the vehicle’s motors are turned off, and the propellers are dismounted.

### Experimental Data Validation with Semi-Empirical Single-Point Frequency Spectrum Models

The semi-empirical single-point frequency spectrum models of Efimtsov [31], Laganelli [14], Lowson [15], Goody [32], Robertson [33] and Smol’yakov [12] are considered here. They have been developed partially using data from either wind tunnel experiments with flat plates or flight tests. They assume a fully developed and attached TBL flow with zero-mean pressure gradient. As a result, the flow can be considered stationary and homogeneous in the plane of the wall. A further assumption is that the aerodynamic excitation can be treated in isolation from the structure [26]. The models are evaluated and compared to the PSD of the microphones’ signals recorded during the wind tunnel experiments in the range of 1 to 10 kHz. The reason for choosing the experimental data from the wind tunnel is that there is accurate information available about the environmental conditions (air density, dynamic and absolute pressure and temperature) coming from sensors in the wind tunnel’s chamber. Their values are required in order to evaluate the single-point frequency spectrum models. Furthermore, these models do not account for the contributions of a vehicle’s engines or motors to the predicted frequency spectrum, making the wind tunnel data corresponding to motor configuration D the most suitable experimental dataset to compare against. Another limitation of these models stems from the fact that they do not consider the AoA as an independent parameter.

In order to determine how well the single-point spectrum models are able to predict the PSDs of the microphones’ signals from the wind tunnel dataset, the Root-Mean-Square Error (RMSE) between them is calculated. Based on that metric, it is found that Robertson’s model is able to most closely predict the wind tunnel PSDs, with the lowest average RMSE of 6.558 dB*_SPL_* over all airspeed values, achieved for the left-mounted microphone, realized for 0° AoA. At the same time, the model of Goody best predicts the PSD decay rate. Robertson’s model is defined as [33]:
(6)$$\Phi \left(f\right)=\frac{q_{\infty }^{2}\left(\frac{\bar{P^{2}}}{q_{\infty }^{2}}\right)}{\omega_{0}{\left(1+{\left(\frac{2\pi f}{\omega_{0}}\right)}^{0.9}\right)}^{2}}$$
with:P2¯q∞2=(0.006)2(1+0.14M2)2,ω0=U∞2δ*,δ*=δ[1.3+0.43M2]10.4+0.5M2[1+2×10−8Rex]1/3andδ=0.37xRex1/51+Rex6.9×10721/10
where $q_{\infty }$ is the dynamic pressure, $\bar{P^{2}}$—mean-square pressure, *M*—Mach number of the airflow, $\omega_{0}$—angular frequency, $U_{\infty }$—free-stream velocity, δ—boundary layer thickness, $\delta^{*}$—displacement thickness and Rex—Reynolds number based on distance.

Figure 9 shows the wind tunnel PSDs with magnitude expressed in dB*_SPL_* for motor configuration D and 0° AoA, together with the prediction of Robertson’s model for the same environmental conditions as those measured in the wind tunnel chamber, namely temperature of 22 °C and air density of 1.194 kg/m^3^. The dynamic viscosity of air is taken to be $\mu =1.837\times 10^{-5}$ kg/(m·s). It is observed that the similarity between the PSDs predicted by the semi-empirical model and the experimental PSDs decreases as the airspeed increases, which is also confirmed by the corresponding RMSE for each airspeed value shown in Table 1. The significant dissimilarity between the spectra predicted by the semi-empirical single-point frequency spectrum models and those of the experimental data provided by the microphones could be attributed to the scaling of the models to the flow conditions of interest. Other researchers report similar discrepancies in data collected from flush-mounted microphones on a flat plate placed in a wind tunnel [26]. They identified the absence of a sizable overlap region, caused by an under-developed logarithmic region of the boundary layer, as the potential root cause for the discrepancies between their microphones’ measurements and the models’ predictions for frequencies above 0.8 kHz. In conclusion, the resulting large error between the single-point frequency spectrum models’ predictions and the microphones’ measurements means that PSDs predicted by these models cannot be used to develop a relation that can be used to estimate the airspeed of an actual vehicle in-flight with satisfactory accuracy.

## 5. Modeling

To model the relationship between the PSD of the microphones’ signals and the airspeed, a supervised learning approach is applied that tackles a parametric nonlinear regression problem, where the independent variable is the PSD of the microphones’ signals and the dependent on the airspeed of the vehicle. To construct the model, the PSDs of the microphones’ voltage signals are used, expressed in dB. This is done to reduce the computational load of processing the signals and computing the PSD with the aim of achieving real-time operation of the airspeed instrument. The downside of not expressing the signals in terms of the pressure is that they become dependent on the microphone’s sensitivity, meaning that the constructed models could only be utilized if the airspeed instrument is fitted with microphones of similar sensitivity.

The chosen model structure is a feed-forward Artificial Neural Network (ANN), as it can be used for any kind of input-output mapping, making it a suitable choice for the problem at hand. It has one hidden layer, allowing it to fit any finite input–output mapping problem, given there are enough hidden neurons in the layer [34]. The ANN is trained with the relevant frequency range (1 kHz until 10 kHz) from the PSDs obtained from 1 s audio data segments as input and the corresponding airspeed as the desired output. The training dataset is made out of all of the wind tunnel data, and 75% of the data from each of the two flight tests. The remaining 25% of the data from both flights constitute the validation dataset. The indices of the flight data are randomized prior to dividing them with the aim of obtaining more diverse data for the validation dataset. An ANN is trained for each of the different types of data collected from the performed experiments—signals from the left and right microphones, as well as the subtracted signals. The activation function of the hidden neurons is chosen to be a hyperbolic tangent, as it was found to yield a lower RMSE over the validation set, compared to other suitable activation functions such as the Sigmoid, Gaussian, or rectified linear unit.

The training is carried out with a second-order gradient descent method that takes into account the local geometry of the function, thus providing a more global view and a higher probability of finding a global minimum. The Levenberg–Marquardt algorithm was chosen, which adaptively varies the ANN weight updates between the first-order steepest descent and the Gauss–Newton method, thus converging faster compared to first-order algorithms [35].

The initialization of the weights and biases of the feed-forward ANN has a significant impact on the training process and the resulting model accuracy. To initialize these parameters, the Nguyen–Widrow algorithm is used. It generates the initial values of the weights and biases such that active regions of the layer neurons will be distributed evenly over input space [36]. The weights at the hidden layer are randomly selected in the interval [−1,1] and scaled according to w=ρw/||w||, where *w* is the input weight vector, and ρ is the magnification factor given as $\rho =0.7H^{1/N}$ with *H* being the number of hidden neurons and *N*—the number of input neurons. The weights connecting the hidden neurons to the output neurons are initialized with small random values over the interval [−0.5,0.5] [37]. As a result, the training is sped up due to setting the initial weights of the hidden layer such that each neuron in it is assigned its own interval at the start of the training [38].

Lastly, the number of neurons in the hidden layer needs to be chosen. A lower number of hidden neurons will prevent overfitting to the training data, thus improving the ability of the network to generalize, and it will decrease the time it takes to complete the training. However, having too few hidden neurons will lead to underfitting, which again worsens the accuracy of the trained models. At the same time, having too many neurons in the hidden layer will lead to overfitting, characterized by a continuous increase in the validation error with a continuous decrease in the training error [39]. In order to determine the optimal number of neurons in the hidden layer, a recursive algorithm based on a successive addition of nodes is used. A maximum allowed number of hidden neurons is set equal to the number of input neurons of the network [40]. The algorithm is initialized with $H^{0}$ = 65 neurons, which is 30% of the size of the network’s input layer and during each iteration ΔH=3, hidden neurons are added until the maximum allowed number of hidden neurons is reached or overfitting behavior is exhibited. Using the algorithm, it was found that a network with 140 hidden neurons yields the most optimal performance.

## 6. Results

The wind tunnel data, part of the input–output dataset utilized when training the feed-forward ANN model structures, only contains data corresponding to an AoA of up to 5° for each motor configuration. The reason behind limiting the wind tunnel datasets to AoA of only up to 5° is that it was found that the models trained on this wind tunnel data alone, next to the data from the flight experiments, achieved the best prediction accuracy over the validation dataset. Based on simulated runs performed using MATLAB, the average time for signal processing and inference is obtained. Extrapolating to the clock frequency of the Teensy 4.1’s microprocessor, the expected signal processing and inference time during operations would be approximately 0.09 s.

In total, five models are trained with the aim of selecting a set of initial hyperparamters giving a model with better performance. All training processes are initialized using the Nguyen–Widrow algorithm. It was found that models constructed with the PSD data of the left-mounted microphone have lower average RMSE compared to those constructed with the PSD data of the right-mounted microphone or the PSD data of the subtracted microphones’ signals. The models trained with the PSD data of the left-mounted microphone had an average RMSE of 1.091 m/s. Among those, the best model has an RMSE of 1.037 m/s. Its mean approximation error is 0.043 m/s with a standard deviation of 1.039 m/s.

For this particular model, Figure 10 depicts the approximation errors between the model’s predictions and the observations given by the validation dataset, as well as the model’s prediction accuracy expressed in percentages using box plots. The central line inside the box represents the median, while the top and bottom sides of the box indicate the 25th and 75th percentiles, respectively. The whiskers extend to the most extreme data points that are not considered outliers, while the outliers are depicted by the red ‘+’ symbol. The plot on the left-hand side relates to the achieved approximation error, where the median is found to be −0.0396 m/s, and in total, 15 predictions are marked as outliers, which is 9.3% of the validation dataset. The plot on the right-hand side shows the model’s prediction accuracy, where the median is found to be 96.0%, and in total, 12 predictions are marked as outliers, equating to 7.5% of the validation dataset.

Figure 11 depicts the airspeed measured by the Pitot tube during flight and the corresponding prediction by the trained model discussed above. It should be noted that the figure shows how well the instrument can mimic the Pitot tube, which itself is not a perfect sensor, which can explain part of the variance. It can be seen that the general trend of the Pitot tube is captured well with the new measurement method without bias. A particular outlier is the one at 3 m/s airspeed, as predicted by the microphone instrument, where the Pitot tube measured a much higher airspeed. The minimum airspeed in the model’s training data is 6 m/s, so a prediction this low is unexpected and is considered an outlier. The $R^{2}$ value of the airspeed predictions by the trained model with respect to line $U_{Pitot}=U_{Model}$ is found to be $R^{2}=0.922$.

### Effects of Changing Angle of Attack

One of the main advantages of the proposed instrument over the Pitot tube is its ability to provide accurate measures of the airspeed for a wide range of AoA. Therefore, the impact of changing AoA on the resulting PSD estimates for the microphone signals is analyzed. Figure 12 depicts the PSDs of the microphone signals recorded during the wind tunnel experiments for motor configuration R, chosen since running motors more closely represent the reality during flight at an airspeed of 10 m/s, which is the highest airspeed spanning all defined values of the AoA that are part of the wind tunnel experiments.

The behavior of the PSD as the AoA is varied and could be explained by a combination of factors, such as the impact of the nose cone’s geometry upon the local flow velocity and dynamic pressure over its top side [41], boundary layer flow separation with a wake forming beneath it [32], and areas of lower dynamic pressure forming over the top side of the nose cone where the microphones are mounted at high angles of attack. It seems plausible that above 45 degrees AoA, the flow separates upstream of the microphones, as the signal power suddenly drops significantly.

In order to judge the capability of the proposed instrument to predict the airspeed at a wide range of AoA, a model is created, utilizing feed-forward ANN with the same hyper-parameters as those described in Section 5, but having an additional input in the form of the AoA. In an operational scenario, this would correspond to having an AoA sensor mounted on the vehicle’s body that would communicate this data to the airspeed instrument. Due to the AoA being measured only during the wind tunnel testing, currently only data gathered from those experiments can be used to construct the models outlined here. The wind tunnel data for all motor configurations are collected and compiled into a single dataset, which is then divided into training and validation datasets according to the 75%/25% division, respectively. Data indices are again randomized prior to dividing them among the two datasets.

The training procedure is repeated a total of five times in order to find a set of initial hyperparameters that results in a trained model with better accuracy. Every training process is initialized using the Nguyen–Widrow algorithm. The models are again constructed with data from the left-mounted microphone. The average RMSE of the resulting models over the validation dataset is 0.159 m/s. Among the trained models, the best one has an RMSE of 0.152 m/s. In order to illustrate the work of an imperfect real-world AoA sensor, a randomly sampled uncertainty ranging from ±15% is applied to each sample point of the AoA, part of the validation input data. Taking the best model, its RMSE becomes 0.364 m/s. The mean approximation error is 0.027 m/s with a standard deviation of 0.486 m/s.

Figure 13 shows the approximation errors between the model’s predictions and the observations given by the validation dataset, with the uncertainty applied to the AoA validation inputs. The box plot on the left-hand side corresponds to the model’s approximation error, where its median is found to be $4.433\times 10^{-4}$ m/s, while 513 model predictions are marked as outliers, which is 13.2% of the validation dataset. The box plot on the right-hand side corresponds to the model’s prediction accuracy. Its median is at 98.9%, and in total, 603 predictions are marked as outliers, equating to 15.5% of the validation dataset. The reason for having significantly more outliers compared to what is shown in Figure 10 is the shorter interquartile range.

## 7. Conclusions

This paper proposed a novel design for an airspeed instrument aimed at MAVs that utilizes two flush-mounted microphones on the vehicle’s nose cone used to capture the TBL-induced pseudo-sound. The microphones’ configuration was chosen with the aim of performing microphones’ signals subtraction, which would suppress acoustic sounds captured by the microphones. However, the pseudo-sound due to the flow structures in the TBL was dominant over any other sound sources in the microphones’ recordings, resulting in the microphones’ signal subtraction having minimal benefits. Neural networks were trained to relate the PSD of the microphones’ signal to the airspeed measured by the Pitot tube installed in the wind tunnel. It was found that models trained on the PSDs of the left-mounted microphone’s signal alone achieved the lowest approximation error over the validation dataset, lower than those trained on the PSDs of the subtracted signals, which is a further argument for the lack of need for implementing the signal subtraction method. The best model achieved an RMSE of 1.037 m/s with a mean approximation error of 0.043 m/s and a standard deviation of 1.039 m/s. Next to that, models including the AoA as an additional input were trained. Here, only PSD data from the wind tunnel experiments was used as the AoA was recorded. An uncertainty of up to 15% was added to the AoA inputs that were part of the validation dataset to simulate the performance of a non-perfect sensor. The resulting best model had an RMSE of 0.364 m/s and a mean approximation error of 0.027 m/s with a standard deviation of 0.486 m/s. From these results, it could be concluded that the utilization of the proposed instrument concept for estimating the airspeed of a MAV within a wide range of angles of attack is feasible, thus substantiating its further development.

Further development and optimization of the instrument’s functionality could bring it to a technology demonstration level, and more research is required before the instrument can be safely incorporated into operating vehicles. Firstly, the microphones mounting configuration could be changed to placing the microphones behind one another at different distances downstream, as the signals subtraction method yields little to no benefits. This configuration can be used to detect boundary layer flow separation or separation bubbles. Alternatively, the number of microphones can be reduced to just one. However, having two microphones, one behind the other, might be useful to detect the separation of the flow and/or the AoA—added as a recommendation to keep the number of microphones the same but place them behind one another. This has the benefit of decreasing the amount of data that needs to be read and processed by the micro-controller. Secondly, new flight experiments could be carried out with a dedicated AoA sensor installed on the vehicle, allowing to collect data about this parameter during flight. Next to that, an additional airspeed instrument, such as a hot-wire anemometer, could be installed to validate the collected Pitot tube’s measurements. Furthermore, as the hot-wire anemometer does not suffer from reduced accuracy at lower airspeeds, it can also be used to calibrate the Pitot tube at lower airspeeds. Lastly, experiments to study the effects of air density, temperature and humidity on the resulting PSD of the microphones’ signals and, consequently, on the performance of the airspeed instrument are recommended.

Since the AoA has a distinct impact on the PSD of the microphones’ signals, it could be investigated as to whether the proposed instrument would be able to also predict the vehicle’s AoA in conjunction with the airspeed. This could be achieved by modifying the structure of the model to have two outputs, one being the airspeed, and the other, the AoA, while the inputs would still be the PSD of the microphones’ signals.

## Figures and Tables

**Figure 1 sensors-23-02463-f001:**
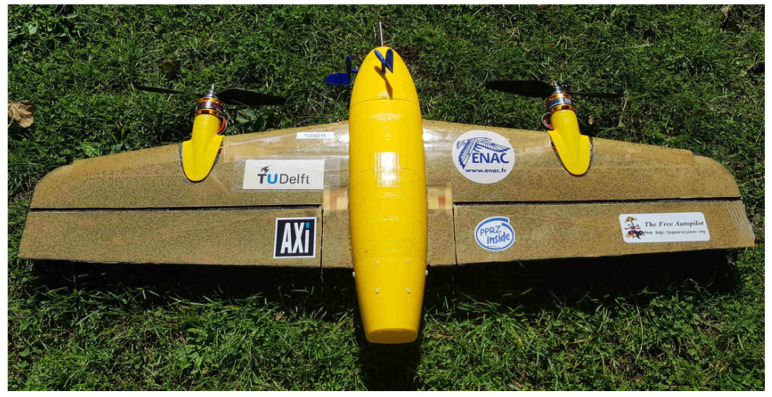
The Cyclone hybrid tail-sitter [19].

**Figure 2 sensors-23-02463-f002:**
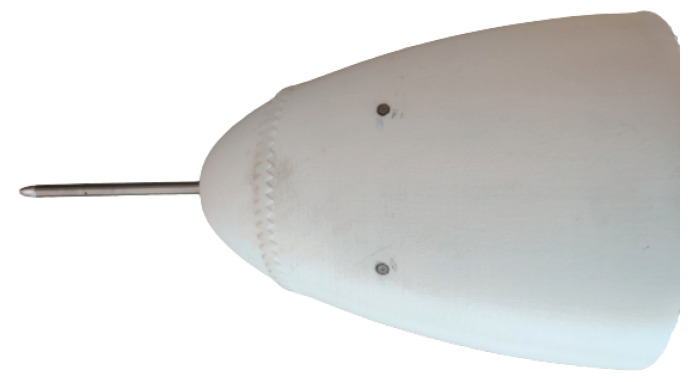
The nose cone of the Cyclone with (from left to right) the Pitot tube, zig-zag tape and the two flush-mounted microphones.

**Figure 3 sensors-23-02463-f003:**
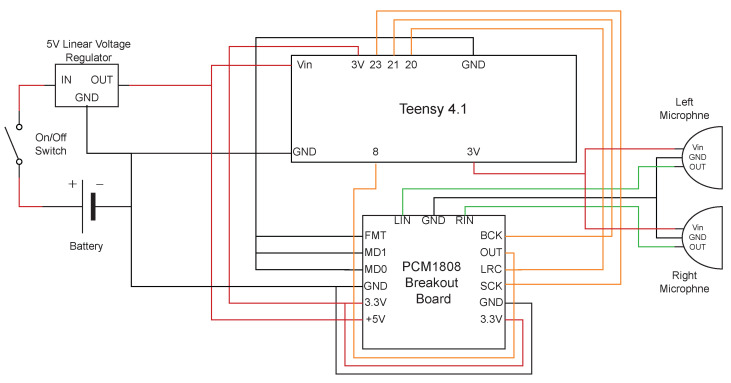
Block diagram of the airspeed instrument. Wires carrying power are colored red, ground wires are black, wires carrying analog signal are green and digital signal wires are orange.

**Figure 4 sensors-23-02463-f004:**
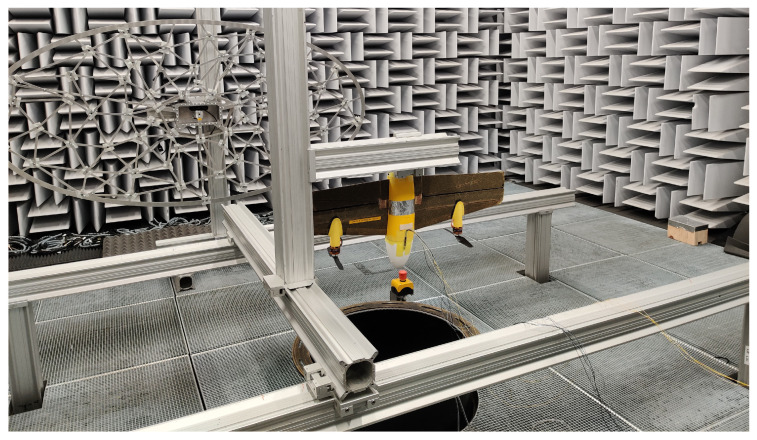
The Cyclone drone mounted on the Cyclone drone mounted on the rig in the A-Tunnel of the TU Delft’s Low-Speed Laboratory.

**Figure 5 sensors-23-02463-f005:**
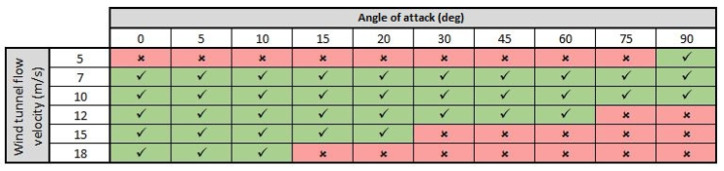
Matrix indicating which experiments were performed in the wind tunnel. Valid for all three motor configurations (D, M, R) defined as part of the experiments. Each cell corresponds to a single experimental run.

**Figure 6 sensors-23-02463-f006:**
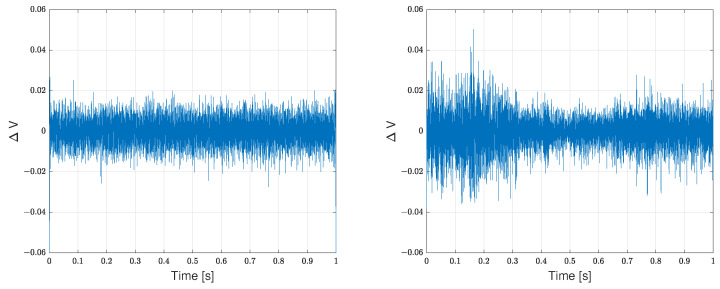
Left microphone data segments from the wind tunnel experimental run at a flow velocity of 12 m/s, AoA of 0° and UAV motor configuration R (**left** plot) and flight data from a phase of steady forward flight at an average airspeed of 12 m/s (**right** plot).

**Figure 7 sensors-23-02463-f007:**
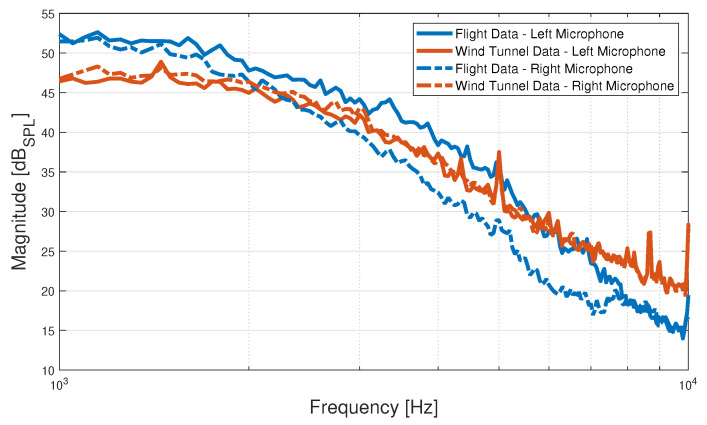
PSDs of the wind tunnel and flight data segments for both the left- and right-mounted microphones at an airspeed of 12 m/s computed using Welch’s method. The wind tunnel data segment corresponds to an AoA of 0° and UAV motor configuration R.

**Figure 8 sensors-23-02463-f008:**
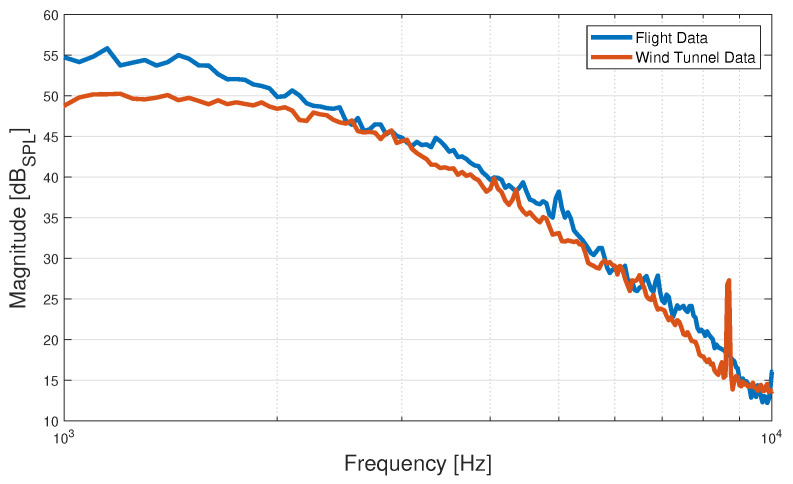
PSDs of wind tunnel and flight data segments for the subtracted microphones signals at an airspeed of 12 m/s, computed using Welch’s method. The wind tunnel data segment corresponds to an AoA of 0° and UAV motor configuration R.

**Figure 9 sensors-23-02463-f009:**
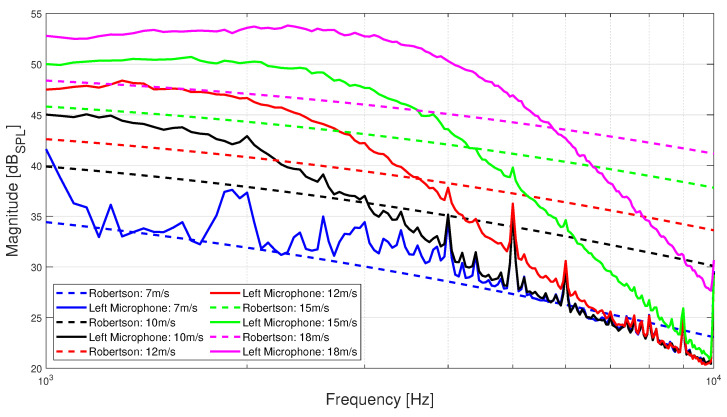
PSDs expressed in dB*_SPL_* of wind tunnel data for the left-mounted microphone wind for motor configuration D and AoA of 0° (solid lines) and those predicted by the Robertson model under the same conditions (dashed lines).

**Figure 10 sensors-23-02463-f010:**
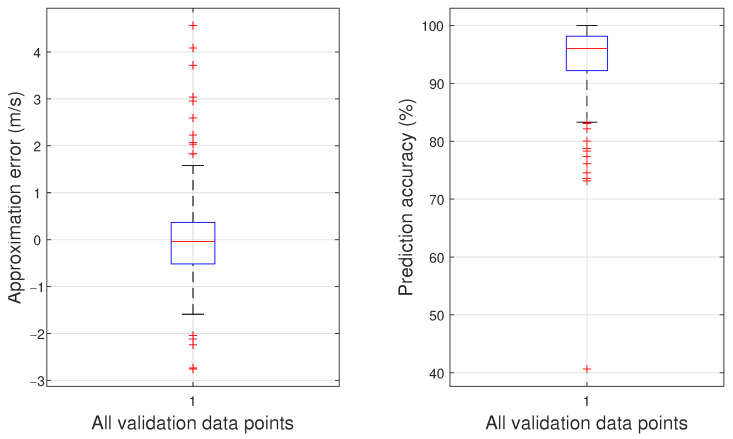
Approximation errors (**left** plot) and prediction accuracy (**right** plot) of the model with the lowest RMSE over the validation dataset.

**Figure 11 sensors-23-02463-f011:**
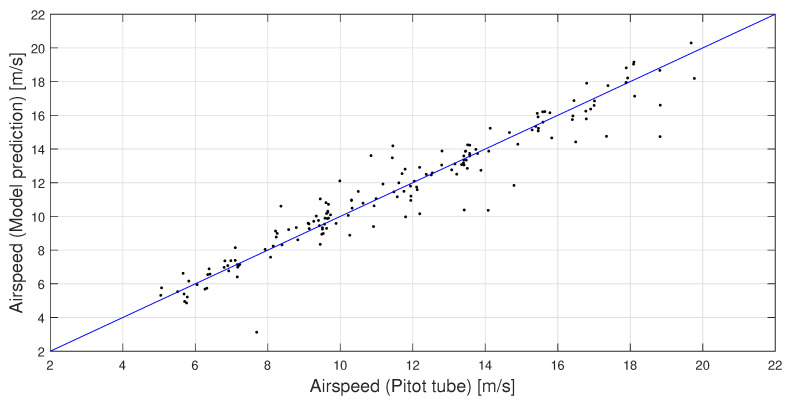
Comparison of the airspeed measured by the Pitot tube and the one predicted by the trained model for the validation dataset.

**Figure 12 sensors-23-02463-f012:**
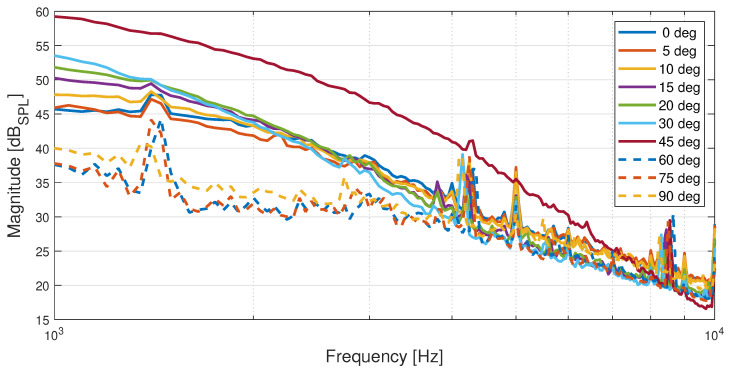
PSDs expressed in dB*_SPL_* of microphone signals recorded as part of the wind tunnel experiments for UAV motor configuration R at an airspeed of 10 m/s for all angles of attack.

**Figure 13 sensors-23-02463-f013:**
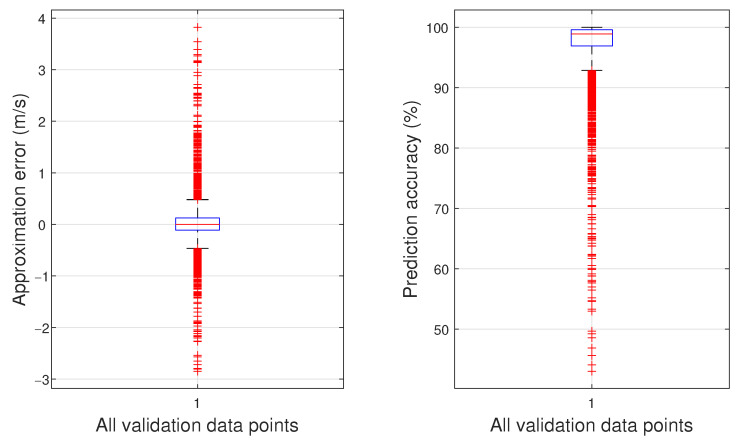
The best model’s approximation errors (**left** plot) and prediction accuracy (**right** plot) over the validation dataset for an uncertainty of up to 15% applied to the AoA model inputs.

**Table 1 sensors-23-02463-t001:** Overall RMSE between the PSDs of the left-mounted microphone data segments from the wind tunnel experiments (motor configuration D and AoA of 0°) and the PSDs predicted by the Robertson model under the same conditions for every airspeed value.

**Airspeed:**	7 m/s	10 m/s	12 m/s	15 m/s	18 m/s
**RMSE:**	2.158 dB*_SPL_*	6.294 dB*_SPL_*	8.388 dB*_SPL_*	9.091 dB*_SPL_*	6.857 dB*_SPL_*

## Data Availability

The research data presented in this study are made openly available by the authors in 4TU.ResearchData at https://doi.org/10.4121/21940367 (accessed on 23 January 2023).

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
