# Peer review of "Microphones as Airspeed Sensors for Unmanned Aerial Vehicles"

_sensors, 2023, doi:10.3390/s23052463_

Round 1

Reviewer 1 Report

The sensitivity of used microphones should be mentioned and their phase characteristics...are the same...model is calibrated in tunnel so it is not so important if the phase is different..

How many averages in obtaining PSD function are used? Why authors don't use the one-third  octave spectrum when analysing the PSD..it add more averaging in the spectrum...maybe the number of coefficients could be smaller in training neural network...

The speed of algorithms, filtering and neural network in determination of speed is necessary to put in the paper..comparison with the other method in determination of speed (GPS system).

What about the influence of meteorological conditions...wind speed on the obtained results (wind speed and propagation speed)  ...if there is some wind speed at higher heights it has the influence on the propagation speed determination. (relative vs absolute value). I don't see the discussion about that...or maybe I missed...

Author Response

We would like to thank the reviewer for their time and effort, and the very qualified comments. We addressed all comments with a change in the manuscript or an explanation. Please observe our detailed response in the attached file.

Reviewer 2 Report

Dear Authors,

I want to thank you for the scientific laboratory work that you presented in the article. Possibility changing the measurements of pilotage navigation parameters on aircraft is hardly possible without a precise methodology that will convince the aviation authorities. Specific UAV-type aerial vehicles are exceptional in that their flight control method is different from the plane itself. The transition of control to telemetric control requires introducing many new techniques that will make the movement of the UAV in the flight area more precise. Also, for this reason, I see your article as innovative when a UAV that moves at low speeds needs to evaluate its flight parameters accurately. Their future use depends on how and to what minimum accuracy we can refine their speeds, flight heights and stabilization with the help of new technologies.

Even so, I have some reservations about the article, which would be appropriate to remove so that your achieved results are clear to the aviation scientific community.

The abstract of the work presented by the authors simply but judiciously shows that the measurements are their own and it is obvious that the reader should proceed with a professional interest in analyzing it.

The Introduction is simply described, also for the reason that the professional content is clear to everyone. On board the aircraft, there are only a few physical parameters with which it is possible to work on their changes without changing flight safety. The aviation public, where every inaccuracy leads to aircraft accidents sensitively watched these changes. This includes measurements of height, pressure, speed and direction on the flight. All piloting and navigation information are currently being addressed as issues for UAV vehicles. As a reason, we take into account the issue of telemetric control and control of UAV vehicles, where the direct measurement of these parameters is delayed. And that’s because of the pilot’s indirect control of the flying vehicle. The manuscript's introduction could sufficiently justify the accuracy of the change in flight speed measurement. Why and what method did, the authors choose and what should be the shift in their established speed measurement that they describe in the theory of flight safety?

The authors prefer a proven fact: this exists, and this is possible. The evaluation in the article is missing, but the professional public cannot deal with this in the next assessment.

As a confirmation of the laboratory observation, the authors chose their own UAV vehicle, which should be described in advance with the existing speeds and the justification for why this particular Cyclon vehicle is suitable for the measurement.

 Other comments that need to be resolved:

 Line - 95. description of the picture is not clear enough about how and where the given microphones are installed on the airframe of the Cyclone laboratory equipment.  How are they structurally situated from a technical point of view? This would also be specified in the technical description on Lines 98-99.

Line 106–110. A very important finding by the authors is the issue of sound capture from the surroundings of the UAV Cyclon. This affects the measurements of the technical environment of the UAV by their chosen sensors (microphones). It is appropriate for the authors to be able to define the entire frequency spectrum that is around the UAV as disturbing. In the case of mechanical undulations (represented by the frequency), I consider it necessary that this problem also be presented in an appropriate form in the article. Of course, it would be appropriate to measure it in an attenuation chamber. But in the next one, the authors describe it suitably.

The authors solve the identified problem in detail in the article, but it would be appropriate to emphasize it for the readers at the beginning of this subchapter.

Line - 140. In Chapter 2.2, the instrument panel (dashboard) assembly with the prerequisite for measuring and evaluating the laboratory experiment is directly described. I ask the authors to write in the introduction of this subchapter a few introductory sentences about what the dashboard solves and how it was selected for the above descriptions in the article.

Line – 154. In Figure 3 in PCM 1880 between LIN and RIN there is (in black) the signal to both microphones. It is not clear from the figure what is the value or data presented by a negative value, or the resulting control variable.

Line 189. In the given picture no. 5, the name "angle of attack" needs to be corrected, please correct it.

Line 254. Put the Pa value in parentheses or modify it according to the template.

Line 258. I see the description of figure 7 as very confusing and take it as a key figure. I ask that the presented key different values be better presented.

Line - 301. Is the definition of Robertson's model (6) presented, where in the article is its implementation in the form of a graph or its use for the purpose of evaluation? It is necessary to add it to Line 313.

Line - 400. Please clarify the dispersion with an indication of the difference between the pitot tube and the model. The regression of variance is indicated in blue, but I can't explain it further. I'm asking for a supplement.

Conclusion: I consider the article to be scientific, which has its value in the way of measuring the speed of flight on other means of flight than the aircraft itself with the help of pitot tubes. The specification of UAVs and their introduction into general aviation activities results in scientific authorities handling the use of other physical methods that would refine pilot navigation information for their safe flight. At the same time, I must indicate that the article is difficult to read for non-aeronautical experts who do not see the differences in the mentioned measured quantities and their problems on the planes themselves during the flight.

However, it is necessary for the authors to reevaluate the article in the user's area of tables, where the article is interspersed with many numerical data that are very difficult to read for an ordinary interested reader who is seeking inspiration in the measurement methodology presented by the authors.

And that hires Lines 302–325, chapter 6 (Results) numerous data in the lines where the reader gets lost, he often has to navigate the article with difficulty others. I will be happy if the authors take it into account.

Author Response

We would like to thank the reviewer for their extensive review and their time and effort, and the very qualified comments. We addressed all comments with a change in the manuscript or an explanation. Please observe our point-by-point response in the attached file.

Reviewer 3 Report

The above manuscript is up to the mark that the main thing I recommend to accept in its present form. 

Author Response

We would like to thank the reviewer for their time and effort, and their recommendation of acceptance.

Reviewer 4 Report

The manuscript ‘Microphones as Airspeed Sensors for Unmanned Aerial

Vehicles mainly focus on putting forward a novel design for an airspeed instrument aimed at small fixed-wing tail-sitter unmanned aerial vehicles.

Remark I: The authors have trained the neural network using data obtained from wind tunnel and flight experiments but have not given an in-depth explanation about the training model. This part needs to be improved.

Remark II: The initialization of the training process needs more explanation. The authors have mentioned the use of the Nguyen-Widrow initialization algorithm for this purpose but have not presented supporting data to depict the process.

Plots and figures can be improved. Especially the Figure 3. Block diagram of the airspeed instrument.

After the above-mentioned explanations are given, the manuscript can be considered for publication

Author Response

We would like to thank the reviewer for their time and effort, and the very qualified comments. We addressed all comments with a change in the manuscript or an explanation. Please observe our point-by-point response in the attached file.

Round 2

Reviewer 2 Report

Dear Authors,

I want to thank you for the explanation and inclusion of partial review notes. I must state that your article has a scientific and professional level in the new method of measuring small UAV flight airspeeds. I believe that the incorporated technology will be an inspiration for the construction and accurate detection of speed within the movement of the UAV even in difficult flight conditions.

After the submitted edits, I recommend the article for publication.